# Tuning the selectivity of NH$_3$ oxidation via cooperative electronic interactions between platinum and copper sites

Lu Chen [1,8] ✉, Xuze Guan[1,8], Zhaofu Fei [2], Hiroyuki Asakura [3], Lun Zhang[1], Zhipeng Wang[1], Xinlian Su [1], Zhangyi Yao[1], Luke L. Keenan [4], Shusaku Hayama[4], Matthijs A. van Spronsen [4], Burcu Karagoz[4], Georg Held [4], Christopher S. Allen [5,6], David G. Hopkinson[5], Donato Decarolis[4,7], June Callison[7], Paul J. Dyson [2] ✉ & Feng Ryan Wang [1] ✉

Selective catalytic oxidation (SCO) of NH$_3$ to N$_2$ is one of the most effective methods used to eliminate NH$_3$ emissions. However, achieving high conversion over a wide operating temperature range while avoiding over-oxidation to NO$_x$ remains a significant challenge. Here, we report a bi-metallic surficial catalyst (Pt$_S$CuO/Al$_2$O$_3$) with improved Pt atom efficiency that overcomes the limitations of current catalysts. It achieves full NH$_3$ conversion at 250 °C with a weight hourly space velocity of 600 ml NH$_3$·h$^{-1}$·g$^{-1}$, which is 50 °C lower than commercial Pt/Al$_2$O$_3$, and maintains high N$_2$ selectivity through a wide temperature window. *Operando* XAFS studies reveal that the surface Pt atoms in Pt$_S$CuO/Al$_2$O$_3$ enhance the redox properties of the Cu species, thus accelerating the Cu$^{2+}$ reduction rate and improving the rate of the NH$_3$-SCO reaction. Moreover, a synergistic effect between Pt and Cu sites in Pt$_S$CuO/Al$_2$O$_3$ contributes to the high selectivity by facilitating internal selective catalytic reduction.

The global emissions of ammonia (NH$_3$) from vehicle exhaust and industrial waste gas streams are estimated to exceed 220,000 tonnes per year, presenting a severe environmental threat and impacting on human health[1,2]. With the utilization of NH$_3$ as a fuel for various modes of transportation, emissions might increase considerably[3,4]. Consequently, selective catalytic oxidation (SCO) of NH$_3$ to nitrogen N$_2$ (avoiding over-oxidation to NO$_x$) is increasingly imperative in order to address the issue of unreacted NH$_3$ emissions (referred to as NH$_3$ slip). NH$_3$-SCO offers an attractive approach for treating gas flows containing 100−5000 ppm NH$_3$ and abundant O$_2$, such as waste gases from chemical manufacturing processes, selective catalytic reduction (SCR)

of NO$_x$ units, biomass gasification processes and NH$_3$ combustion turbines.

NH$_3$-SCO catalysts must be capable of achieving complete conversion of NH$_3$ to N$_2$, while avoiding overoxidation to NO$_x$, and maintaining high stability over a broad operating temperature range (100 °C < T < 450 °C). Noble metal-based catalysts, such as commercial Pt/Al$_2$O$_3$, are renowned for their high efficiencies, however, their selectivity to N$_2$ is low, (ca. 50% for commercial Pt/Al$_2$O$_3$) and the atomic efficiency is also low[5]. First row transition metals such as Cu and Fe exhibit high N$_2$ selectivity, but require higher operating temperatures (300−500 °C)[6,7]. In order to enhance the N$_2$ selectivity of Pt-based

[1]Department of Chemical Engineering, University College London, London WC1E 7JE, UK. [2]Institute of Chemical Sciences and Engineering, École Polytechnique Fedérale de Lausanne (EPFL), 1015 Lausanne, Switzerland. [3]Department of Applied Chemistry, Faculty of Science and Engineering, Kindai University, Higashi-Osaka, Osaka 577-8502, Japan. [4]Diamond Light Source Ltd., Harwell Science and Innovation Campus, Chilton, Didcot OX11 0DE, UK. [5]electron Physical Science Imaging Center, Diamond Light Source Ltd., Harwell Science and Innovation Campus, Chilton, Didcot OX11 0DE, UK. [6]Department of Materials, University of Oxford, Oxford OX1 3PH, UK. [7]UK Catalysis Hub, Research Complex at Harwell (RCaH), Rutherford Appleton Laboratory, Harwell OX11 0FA, UK. [8]These authors contributed equally: Lu Chen, Xuze Guan. ✉e-mail: lc962@cam.ac.uk; paul.dyson@epfl.ch; ryan.wang@ucl.ac.uk

catalysts, recent developments have aimed at the development of catalysts that leverage the strengths of noble metals and non-noble transition metals through bifunctional catalyst design. Efforts have focused on the bi-metallic catalysts that form somewhat random alloys or mixed oxides[8–16], and the location of noble metal atoms is difficult to control. However, noble metals nanoparticles favour the oxidation of ammonia to NO, whereas non-noble transition metal oxides are able to selectively reduce NO with $NH_3$ to generate $N_2$[7–9]. Therefore, by combining the two types of catalysts using a rational design approach, i.e. ensuring the ratio of the Pt and CuO is optimized to the relative reaction rates of both processes, it should be possible to obtain a highly active and selective catalyst for the $NH_3$-SCO reaction.

Herein, by precisely controlling the amount of Pt atoms on the surface (S) of CuO nanoparticles (NPs, $Pt_SCuO/Al_2O_3$), the catalytic activity and $N_2$ selectivity were enhanced, surpassing commercial Pt/$Al_2O_3$ catalysts and also standard co-reduced bi-metallic $Pt_NCuO/Al_2O_3$ (N stands for normal) catalysts. Based on *operando* and time-resolved X-ray absorption fine structure (XAFS) studies, the Pt atoms are active in the oxidation of $NH_3$. Moreover, the Pt atoms also accelerate the redox activity of the Cu species facilitating the $NH_3$-SCO reaction to achieve high activity and high selectivity to $N_2$.

## Results and Discussion
### Synthesis and structural characterisation of the $Pt_SCuO/Al_2O_3$ catalyst
The surface bi-metallic $Pt_SCuO/Al_2O_3$ catalyst was synthesized using galvanic replacement. As the galvanic replacement between Pt and Cu atoms is initiated on the CuO NPs, the Pt atoms form a thin shell on the NP surfaces. For comparison, normal alloy $Pt_NCuO/Al_2O_3$ with the same chemical compositions was synthesized by wet impregnation, resulting in randomly distributed Pt atoms within the CuO NPs. The NPs in both $Pt_SCuO/Al_2O_3$ and $Pt_NCuO/Al_2O_3$ have a similar average particle size of ca. 2 nm (Fig. 1a, b, Supplementary Figs. S1 and S2). The

Energy-dispersive spectrometry (EDS) map of $Pt_SCuO/Al_2O_3$ displays a uniform elemental distribution over a wide area (Fig. S3), and within a single $Pt_SCuO/Al_2O_3$ particle (Fig. S4). X-ray diffraction patterns of $Pt_SCuO/Al_2O_3$ and $Pt_NCuO/Al_2O_3$ shows no obvious distinctions from the $Al_2O_3$ support (PDF #10-0425) (Fig. S5), indicative of small NPs. Fine-scanned X-ray photoelectron spectroscopy (XPS) of $Pt_SCuO/Al_2O_3$ and $Pt_NCuO/Al_2O_3$ reveals Cu $2p_{3/2}$ peaks at 931.8 and 934.6 eV corresponding to $Cu^+$ and $Cu^{2+}$, respectively (Fig. S6). The chemical environments of the copper ions in both $Pt_SCuO/Al_2O_3$ and $Pt_NCuO/Al_2O_3$ are similar, as determined from extended X-ray absorption fine structure (EXAFS) and X-ray absorption near-edge structure (XANES) measurements, but differ in the chemical states of Pt (Fig. 1c, d and S7). The fitting of EXAFS data reveals the differences between the structures of $Pt_SCuO/Al_2O_3$ and $Pt_NCuO/Al_2O_3$ (Table S1). $Pt_SCuO/Al_2O_3$ has a larger Pt-Pt coordination number (C.N.) of $5.8 \pm 1.5$ and smaller Pt-Cu C.N. of $1.1 \pm 0.7$ compared to $Pt_NCuO/Al_2O_3$, which has a Pt-Pt C.N. of $4.5 \pm 1.4$ and a Pt-Cu C.N. of $5.8 \pm 1.7$. A smaller overall coordination indicates that most Pt atoms in the $Pt_SCuO/Al_2O_3$ are on the surface whereas the Pt atoms in $Pt_NCuO/Al_2O_3$ reside on the surface and in the NP bulk.

### Evaluation of the catalysts in the $NH_3$-SCO reaction
The activities of $Pt_SCuO/Al_2O_3$, $Pt_NCuO/Al_2O_3$, CuO/$Al_2O_3$ and Pt/$Al_2O_3$ were evaluated in the $NH_3$-SCO reaction. Remarkably, at 200 °C, $Pt_SCuO/Al_2O_3$ consisting of 0.6 wt% Pt and 4.4 % Cu, exhibits a 30-fold higher activity than the commercial catalyst Pt/$Al_2O_3$ containing 1 wt% Pt. Moreover, $Pt_SCuO/Al_2O_3$ also shows the highest activity with the lowest $T_{50}$ (T at 50% conversion), and complete $NH_3$ conversion was achieved at around 250 °C (Fig. 2a, S8 and S9). Under realistic $NH_3$ slip conditions (1000 ppm $NH_3$, weight hourly space velocity (WHSV) of 120 $ml_{NH3} \cdot h^{-1} \cdot g^{-1}$), full conversion could be achieved at 200 °C using the $Pt_SCuO/Al_2O_3$ catalyst (Fig. 2a). Notably, $Pt_NCuO/Al_2O_3$ with identical Pt and Cu loadings to $Pt_SCuO/Al_2O_3$, requires 300 °C to reach the

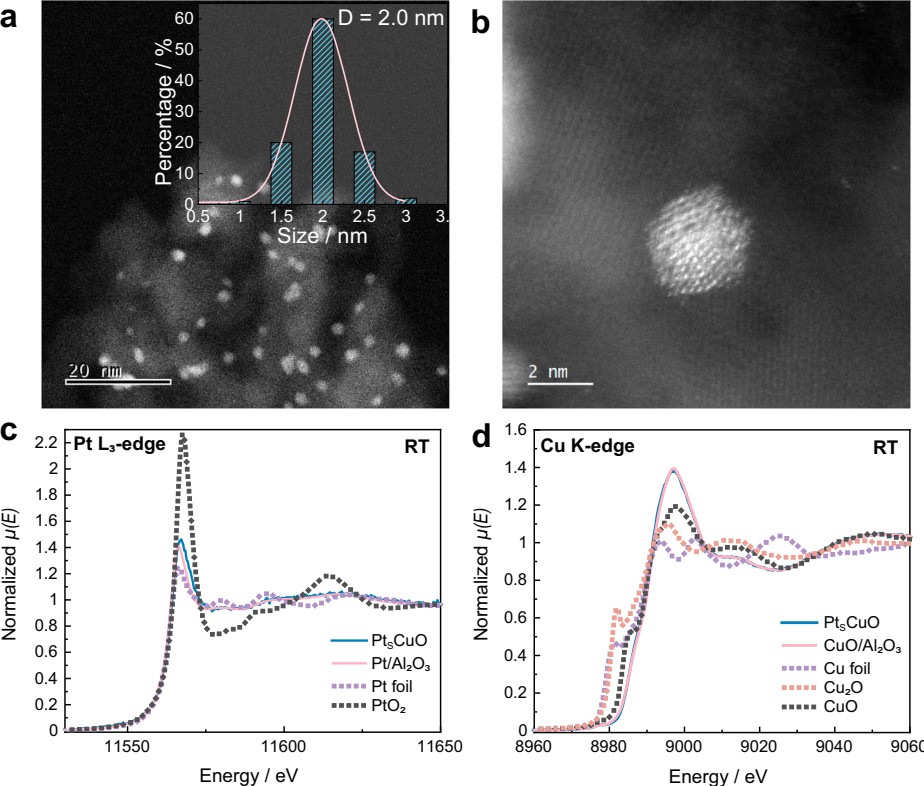

**Fig. 1 | Characterisation of $Pt_SCuO/Al_2O_3$ and $Pt_NCuO/Al_2O_3$. a, b** TEM images of $Pt_SCuO/Al_2O_3$ (the inset in **a** shows the particle size distribution. The average particle size was calculated based on more than 100 particles.); Pt $L_3$-edge and Cu K-edge EXAFS of $Pt_SCuO/Al_2O_3$ (**c**) and $Pt_NCuO/Al_2O_3$ (**d**).

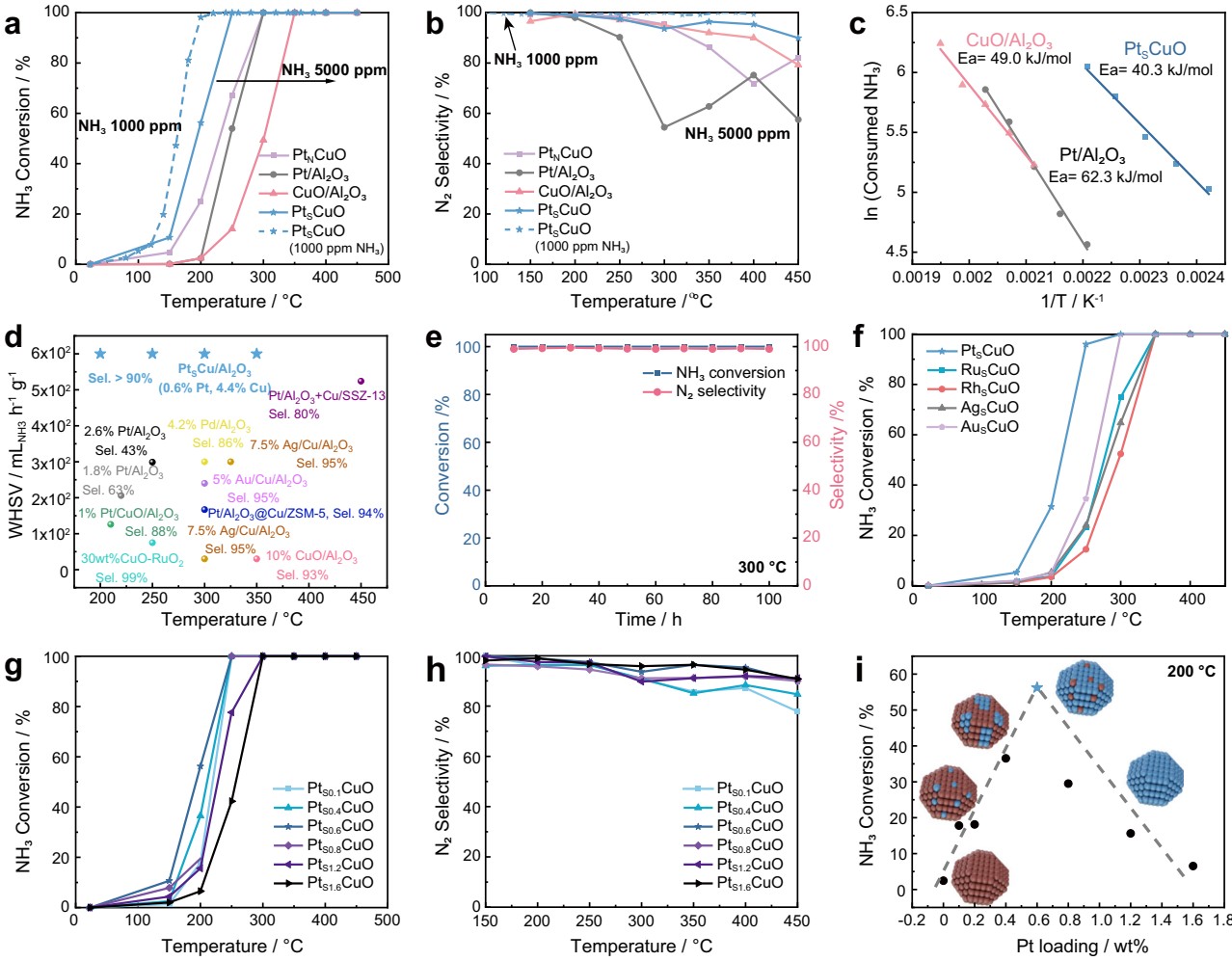

**Fig. 2 | Evaluation of Pt$_S$CuO/Al$_2$O$_3$ in comparison to other catalysts in the NH$_3$-SCO reaction. a, b** NH$_3$ conversion and N$_2$ selectivity as a function of temperature; **c** activation energy of Pt$_S$CuO/Al$_2$O$_3$, CuO/Al$_2$O$_3$ Pt/Al$_2$O$_3$; **d** WSHV with refs (Table S2 [10–23]); **e** stability test of Pt$_S$CuO/Al$_2$O$_3$ at 200 °C; **f** activity of M$_S$CuO/Al$_2$O$_3$ catalysts (M = Pt, Ru, Rh, Ag, or Au); **g, h** NH$_3$ conversion and N$_2$ selectivity as a function of Pt loading and temperature; **i** NH$_3$ conversion of Pt$_S$CuO/Al$_2$O$_3$ with different Pt loadings at 200 °C. Reaction conditions: 50 mg catalyst, 5000 ppm NH$_3$, 5% O$_2$ balanced in He, gas flow: 100 mL/min, WHSV = 600 mL NH$_3$·h$^{-1}$·g$^{-1}$.

full conversion under equivalent conditions (Fig. 2a). When the temperature is higher than 250 °C, the N$_2$ selectivity of the Pt$_N$CuO/Al$_2$O$_3$, CuO/Al$_2$O$_3$ and Pt/Al$_2$O$_3$ catalysts decrease, due to an increase in the rate of ammonia oxidation to NO (Step 1: 4 NH$_3$ + 5O$_2$ → 4 NO + 6 H$_2$O). In comparison, Pt$_S$CuO/Al$_2$O$_3$ consistently maintains >90% selectivity to N$_2$ even at full conversion (Fig. 2b). Pt/Al$_2$O$_3$ has the lowest selectivity to N$_2$ at temperatures above 250 °C, e.g. 55% selectively at 300 °C.

A series of control experiments were carried out to ensure the influence of internal and external diffusion could be excluded from the kinetic experiments (Fig. S10a, b). The apparent activation energy of Pt$_S$CuO/Al$_2$O$_3$ is 40 kJ/mol, which is lower than that of CuO/Al$_2$O$_3$ and Pt/Al$_2$O$_3$ (Fig. 2c), suggesting that the superior activity in the NH$_3$-SCO reaction is due to interactions between the surface Pt atoms and underlying Cu atoms (note that other metal-Cu combinations are less efficient, see below). Additionally, Pt$_S$CuO/Al$_2$O$_3$ outperforms most previously reported catalysts in terms of selectivity to N$_2$ (Fig. 2d and Table S2)[10–23]. Compared to benchmark Pt- and Cu-based catalysts, such as the Pt/Al$_2$O$_3$@Cu/ZSM-5 core−shell catalyst and Pt/Al$_2$O$_3$ + Cu/SSZ−13 dual-layer wash-coated monolith catalyst, Pt$_S$CuO/Al$_2$O$_3$ displays a higher activity at low temperatures and maintains high selectivity at high temperatures[20–23]. Notably, Pt$_S$CuO/Al$_2$O$_3$ also exhibits exceptional stability, without any signs of reduced activity and N$_2$ selectivity even after 100 h of continuous operation at 200 or 300 °C (Fig. 2e and S10c). After the reaction, the size distribution of the NPs in

the Pt$_S$CuO/Al$_2$O$_3$ catalyst remains unchanged (Fig. S11), affirming its remarkable stability. The Pt$_S$CuO/Al$_2$O$_3$ catalyst is able to operate at low temperatures (200 °C) that are well suited to the cold start of a vehicle, and its high N$_2$ selectivity over a wide operating temperature range (150−450 °C) makes it practical for removal of NH$_3$ from diesel exhausts.

As mentioned above, the Pt and Cu combination in Pt$_S$CuO/Al$_2$O$_3$ shows the highest activity, with M$_S$CuO/Al$_2$O$_3$ (M = Pt, Ru, Rh, Ag, or Au) catalysts prepared via galvanic replacement being less active (Fig. 2f and S12). To investigate the special effect of Pt coverage on CuO NPs in the NH$_3$-SCO reaction, the Pt loading was systematically varied from 0 to 1.6 wt% and the performance of the catalysts was evaluated (Fig. 2g–i and S13). Despite all catalysts exhibiting similar particle size distributions irrespective of the Pt loading (Figs. S14−16), their catalytic performance differs significantly. At 200 °C the activity of Pt$_S$CuO/Al$_2$O$_3$ shows a volcano plot-like trend with 0.6 wt% Pt, i.e. Pt$_{S0.6}$CuO/Al$_2$O$_3$, being the most active, indicating that optimal Pt surface coverage is important (Fig. 2i). Hence, the Pt-Pt and Pt-Cu coordination environments play a vital role in the NH$_3$-SCO reaction.

## Formation of Cu$^+$ and Pt$^0$ at steady state
The internal selective catalytic reduction (i-SCR) mechanism is widely accepted with copper-based catalysts[18,24–27], in which ammonia is oxidized to NO$_x$, and then the formed NO$_x$ species react further with NH$_3$

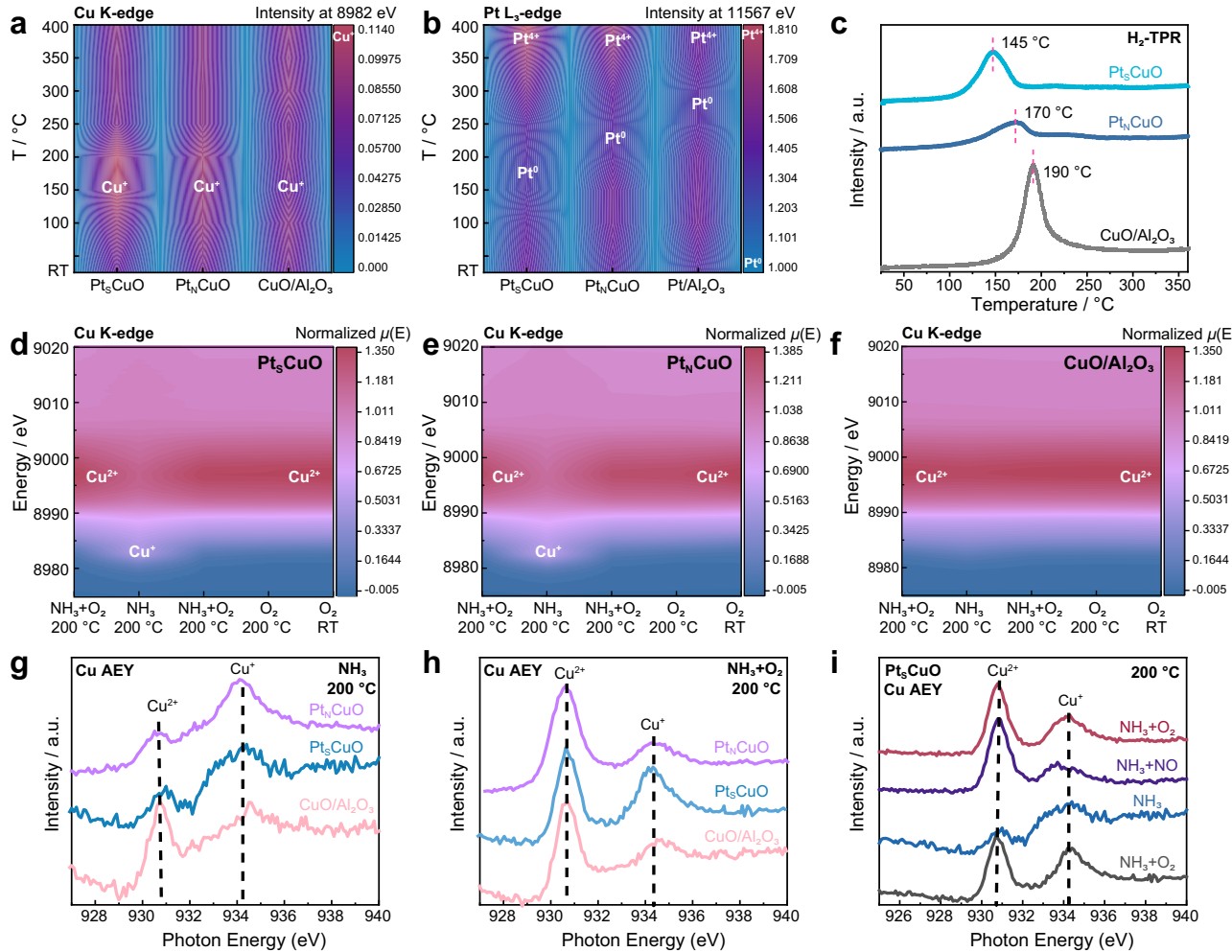

**Fig. 3 | *Operando* studies on the redox behaviour of Cu and Pt in different catalysts. a** *operando* Cu K-edge XAFS, signal intensity of the Cu$^+$ 1s-4p transition peak at 8982 eV in a NH$_3$/O$_2$ atmosphere at different temperatures; **b** *operando* Pt L$_3$-edge, signal intensity of the white line of the Pt L$_3$-edge at 11567 eV in a NH$_3$/O$_2$ atmosphere at different temperatures (reaction conditions: 5000 ppm NH$_3$, 5% O$_2$ balanced in He, gas flow: 100 mL/min); **c** H$_2$-TPR of different catalysts; *operando* Cu K-edge XANES spectra of Pt$_S$Cu/Al$_2$O$_3$ (**d**), Pt$_N$Cu/Al$_2$O$_3$ (**e**) and CuO/Al$_2$O$_3$ (**f**) in different gases at 200 °C; in situ NAP-NEXAFS spectra, Cu L-edge (AEY mode) of different catalysts at 200 °C under NH$_3$ (**g**) or NH$_3$/O$_2$ (**h**) atmospheres; **i** Cu L-edge (AEY mode) of Pt$_S$Cu/Al$_2$O$_3$ under various gas atmospheres at 200 °C (gas pressure 1 mbar).

on the Cu sites to afford N$_2$ (Step 2). The first step is the conversion of NH$_3$ and is usually the rate-determining step in NH3 oxidation, which involves metal redox processes[27].

Step 1: 4 NH$_3$ + 5O$_2$ → 4 NO + 6 H$_2$O
Step 2: 4 NO + 4 NH$_3$ + O$_2$ → 4 N$_2$ + 6 H$_2$O
4 NO + 4 NH$_3$ + 3 O$_2$ → 4 N$_2$O + 6 H$_2$O

Adding Pt into CuO will boost Step 1, as metallic Pt is usually the best catalyst for low-temperature combustion reactions[1,28,29]. In *operando* XAFS under SCO conditions, indicated that the initial Pt$^{4+}$ species in the Pt$_S$CuO/Al$_2$O$_3$, Pt$_N$CuO/Al$_2$O$_3$ and Pt/Al$_2$O$_3$ catalysts are reduced to Pt$^0$ in the temperature ranges 100–250, 200–300 and 250–300 °C, respectively, and are subsequently re-oxidized to Pt$^{4+}$ (Fig. 3b and S17–19). Pt/γ-Al$_2$O$_3$ (2 wt%) has been previously studied by operando XAS and was shown to undergo reduction between 230 – 260 °C[23], which is similar the catalysts reported herein. When the temperature is lower than 250 °C, the Pt species are mainly in metallic state in Pt$_S$CuO/Al$_2$O$_3$, which improved the activity of Step 1 (Figs. 3b, 4g), while Pt species are in a more oxidized state in Pt$_N$CuO/Al$_2$O$_3$, and Pt/Al$_2$O$_3$. In situ, XAFS under different gas atmospheres confirms that Pt species in Pt$_S$CuO/Al$_2$O$_3$ are readily oxidized when the feed gas composition changes from reducing to oxidizing at 200 °C (Fig. S20).

Although Pt is predominant in the surface layer, the underlying Cu atoms modulate the surface electronic structure to improve both the catalytic activity and durability. To assess the redox behaviour (essential for the formation of N$_2$ from the oxidation of NH$_3$) and monitor structural changes of the CuO NPs and surface Pt under relevant NH$_3$-SCO reaction conditions, *operando* XAFS measurements were performed in fluorescence mode at the Pt L$_3$-edge and transmission mode at the Cu K-edge under steady-state conditions between 25 and 400 °C (Fig. 3a–c and S21–23). Figure 3a illustrates the evolution of the Cu K-edge XAFS spectra at the Cu$^+$ 1 s → 4p transition in Pt$_S$CuO/Al$_2$O$_3$, Pt$_N$CuO/Al$_2$O$_3$, and CuO/Al$_2$O$_3$ during the NH$_3$-SCO reaction (NH$_3$ 5000 ppm, O$_2$ 5%) as a function of temperature. Between 100 and 250 °C, Cu$^{2+}$ in Pt$_S$CuO/Al$_2$O$_3$ is partially reduced to Cu$^+$, even under excess O$_2$, as evidenced by the more pronounced peak intensity of the Cu K-edge at 8982 eV (the typical feature for Cu$^I$(NH$_3$)$_2$ 1 s → 4p XANES) (Fig. 3a). Above 250 °C the Cu$^+$ feature at 8982 eV disappears and is accompanied by an increase in the white-line intensity at 8996 eV (Fig. 3a, S21 and 24), indicative of oxidation of Cu$^+$ to Cu$^{2+}$. In comparison, the peak intensity of the Cu K-edge at 8982 eV in Pt$_N$CuO/Al$_2$O$_3$ only increases slightly between 100 and 250 °C, whereas Cu$^+$ in CuO/Al$_2$O$_3$ is almost constant over the entire temperature range (Fig. 3a, S23–25). As the feed gas composition is changed from

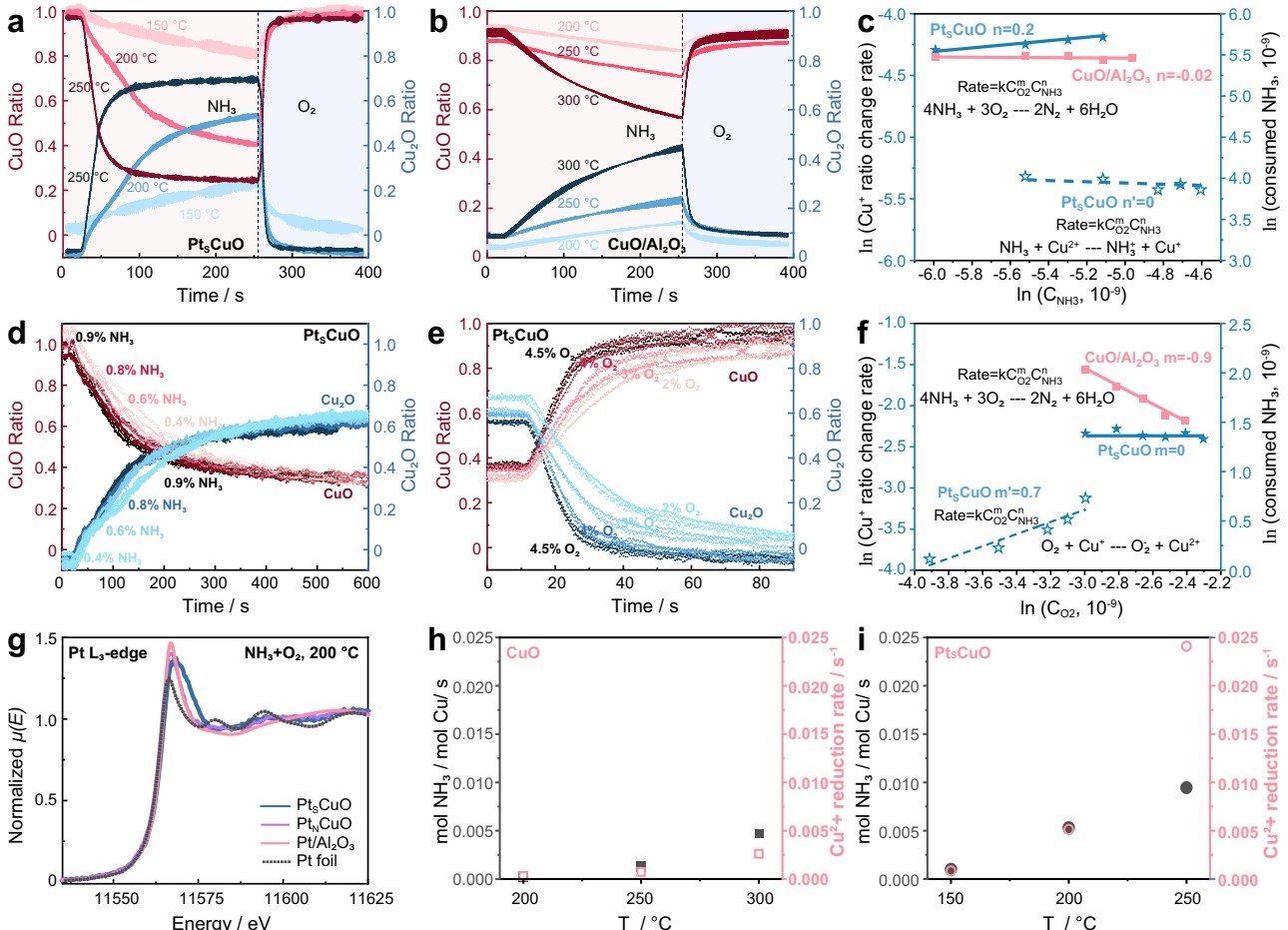

**Fig. 4 | Kinetic studies and adsorption behaviours of the Pt$_S$CuO/Al$_2$O$_3$ and CuO/Al$_2$O$_3$ catalysts.** Change of CuO:Cu$_2$O ratio in Pt$_S$CuO/Al$_2$O$_3$ (**a**) and CuO/ Al$_2$O$_3$ (**b**) with time under NH$_3$ or O$_2$ environments at different temperatures (the catalysts are exposed to a flow of 5000 ppm NH$_3$/He or 5% O$_2$/He with flow rate 15 mL/min); change of CuO:Cu$_2$O ratio in the Pt$_S$CuO/Al$_2$O$_3$ catalyst with time under NH$_3$ (**d**) and O$_2$ (**e**) environments at different gas concentrations; reaction order for NH$_3$ (**c**) and O$_2$ (**f**) in the whole apparent reaction or half reaction; **g** *operando* Cu K-edge XANES spectra of the Pt$_S$Cu/Al$_2$O$_3$, Pt$_N$Cu/Al$_2$O$_3$ and CuO/Al$_2$O$_3$ catalysts under NH$_3$ + O$_2$ at 200 °C (5000 ppm NH$_3$, 5% O$_2$ balanced in He, gas flow: 100 mL/ min); Comparison between TOF of the full NH$_3$-SCO reaction and Cu$^{2+}$ reduction rate in Cu/Al$_2$O$_3$ (**h**) and Pt$_S$Cu/Al$_2$O$_3$ (**i**).

reducing to oxidizing, the Cu species in Pt$_S$CuO/Al$_2$O$_3$ are readily reduced and oxidized at 200 °C (Fig. 3d and S26), whereas the Cu species in CuO/Al$_2$O$_3$ do not change (Fig. 3f).

In situ near ambient pressure-near edge X-ray absorption fine structures (NAP-NEXAFS) of the catalysts under a NH$_3$ atmosphere (1 mbar) confirms the superior redox properties of Pt$_S$CuO/Al$_2$O$_3$, with a more pronounced peak observed at 934.2 eV corresponding to Cu$^+$ (Fig. 3g). Additionally, the formation of Cu$^+$ in Pt$_S$CuO/Al$_2$O$_3$ under a NH$_3$/O$_2$ atmosphere was validated by in situ NAP-NEXAFS (Fig. 3h). In the NAP-NEXAFS studies, the Cu species in the Pt$_S$CuO/Al$_2$O$_3$ catalyst return to their original redox states after exposure to different gases (Fig. 3i). This switching between Cu$^+$ and Cu$^{2+}$ confirms the reversibility of the oxidation state of Cu species in Pt$_S$Cu/Al$_2$O$_3$, as well as the high stability of the catalyst together with the 100 h stability tests at 200 °C (Fig. 2e). Based on the *operando* XAFS and in situ NAP-NEXAFS studies, the Cu species in Pt$_S$CuO/Al$_2$O$_3$ undergo facile redox switching compared to Pt$_N$CuO/Al$_2$O$_3$ and CuO/Al$_2$O$_3$, which is in agreement with the hydrogen-temperature programmed reduction (H$_2$-TPR) results, in which the Pt$_S$CuO/Al$_2$O$_3$ catalyst has the lowest reduction temperature, indicating that surface Pt species enhance the reduction ability of CuO NPs (Fig. 3c).

Reduction of Cu$^{2+}$ to Cu$^+$ and Pt$^{4+}$ to Pt$^0$ takes place in the same temperature range for Pt$_S$CuO/Al$_2$O$_3$, which is not the case for Pt$_N$CuO/ Al$_2$O$_3$, indicating a better redox synergy between the two metals in the former. Hence, it would appear that Pt$^0$ induces the reduction of Cu$^{2+}$ to Cu$^+$ in Pt$_S$CuO/Al$_2$O$_3$. Pt$_N$CuO/Al$_2$O$_3$ and Pt/Al$_2$O$_3$ are more resistant to reduction and oxidation in the course of the reaction, since both the Cu and Pt oxidation states do not vary significantly (Fig. 3a, b). In the light-off curve of Pt$_S$Cu/Al$_2$O$_3$ shown in Fig. S27, the reaction rate increases rapidly from 150 °C, which corresponds to the formation temperature of the Cu$^I$(NH$_3$)$_2$ species (Fig. 3a), so that the formation of the Cu$^+$ is considered to be a trigger for the NH$_3$-SCO reaction. The highest activity for the NH$_3$-SCO reaction between 100 and 250 °C is found for Pt$_S$CuO/Al$_2$O$_3$, which has the highest abundance of Cu$^+$. Compared with Pt$_N$CuO/Al$_2$O$_3$ and CuO/Al$_2$O$_3$, the superior redox properties of Cu and Pt species in Pt$_S$CuO/Al$_2$O$_3$ lead to higher activity at lower temperatures.

### Determination of the Cu$^{+/2+}$ redox rate
In situ energy dispersive EXAFS (EDE) under modulation excitation of net-reducing (NH$_3$) to net-oxidizing (O$_2$) gas environments was used to probe the redox behaviour of Cu$^{+/2+}$ in the Pt$_S$CuO/Al$_2$O$_3$ and CuO/ Al$_2$O$_3$ catalysts. For both catalysts oxidation to Cu$^{2+}$ is much faster than reduction to Cu$^+$, partially due to the higher concentration of O$_2$ (5%) used compared to NH$_3$ (5000 ppm) (Fig. 4a, b). Notably, the rates for oxidation and reduction in Pt$_S$CuO/Al$_2$O$_3$ is much higher than in CuO/ Al$_2$O$_3$ at all temperatures, which is in agreement with the steady-state study (see Fig. 3).

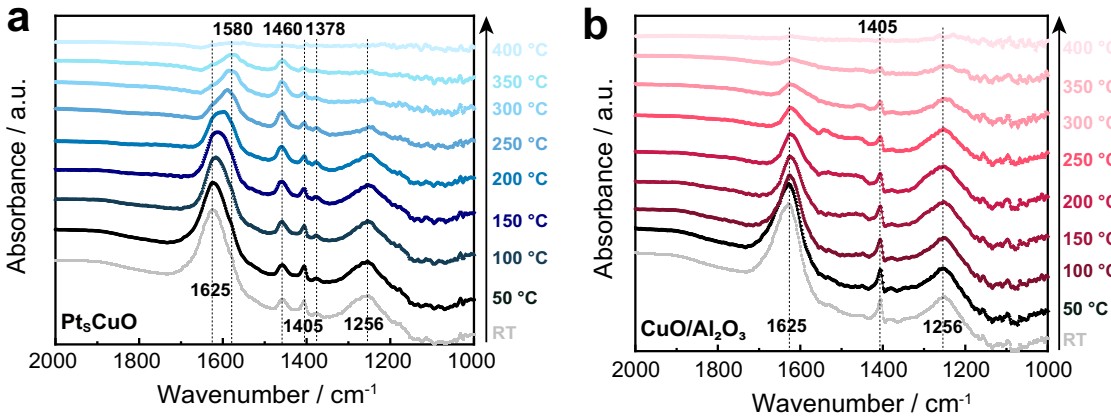

**Fig. 5 | *Operando* DRIFTS studies.** *Operando* DRIFTS spectra of $Pt_SCu/Al_2O_3$ (**a**) and $Cu/Al_2O_3$ (**b**) as a function of temperature (the catalysts were exposed to a flow of 5000 ppm $NH_3$ and 5% $O_2$ for 20 min at different temperatures).

At 200 °C, the reaction order of $NH_3$ in the half-reaction ($2NH_3 + 6CuO \rightarrow N_2 + 3Cu_2O + 3H_2O$) is facilitated by Pt is almost 0, which is consistent with the apparent reaction order of $NH_3$ (0.2) in the overall SCO reaction ($4NH_3 + 3O_2 \rightarrow 2N_2 + 6H_2O$) (Fig. 4c, d). In this case, the $NH_3$ that reacts is limited to those that absorb on the $Pt_SCuO/Al_2O_3$ surface, a key step for both the SCO reaction and the reduction of $Cu^{2+}$. The amount of $Cu^+$ in $Pt_SCuO/Al_2O_3$ is dependent on the oxygen concentration (Fig. 4e, f), with a reaction order of 0.7. The $O_2$ order for the SCO reaction is 0 and −0.9 for the $Pt_SCuO/Al_2O_3$ and $CuO/Al_2O_3$ catalysts (Fig. 4f), respectively, indicating that the lattice O participates in the oxidation process due to the surface coverage of $NH_3$ on the $Pt_SCuO/Al_2O_3$ catalyst (Mars–van Krevelen mechanism)[30,31]. The lower reaction order for $O_2$ in the $NH_3$-SCO reaction implies facile oxygen activation takes place on the surface of the $Pt_SCuO/Al_2O_3$ catalyst. In the absence of $NH_3$, however, for the reaction $O_2 + 2 Cu_2O \rightarrow 4 CuO$, the gas phase oxygen surface coverage is rate determining[27]. Overall, the redox behaviour of $Cu^{+/2+}$ is important for the i-SCR mechanism in the overall $NH_3$-SCO process[12,32], with the $Cu^{+/2+}$ redox kinetics aligned with the overall SCO kinetics.

To further validate that the $Cu^{2+}$ reduction is the rate-determining step, we compare the $Cu^{+/2+}$ redox rate with the turnover frequency (TOF) of the whole $NH_3$-SCO reaction (Fig. 4h, i, S28). With both $Pt_SCuO/Al_2O_3$ and $CuO/Al_2O_3$, the $Cu^+$ oxidation rate is faster than $Cu^{2+}$ reduction rate, and the $Cu^+$ oxidation rate consistently surpasses the $NH_3$ oxidation rate (i.e. TOF of the whole $NH_3$-SCO reaction) (Fig. S28). This implies that the $Cu^+$ oxidation step is not the rate-determining step in both $Pt_SCuO/Al_2O_3$ and $CuO/Al_2O_3$ catalysts. For both catalysts, the TOF is comparable to the rate of $Cu^{2+}$ reduction to $Cu^+$, indicating that $Cu^{2+}$ reduction should be the rate-determining step (Fig. 4h, i). At 250 °C, $NH_3$ conversion for $Pt_SCuO/Al_2O_3$ is over 50% and thus in the diffusion region. Therefore, the $Cu^{2+}$ reduction rate from the in situ EDE experiment is much higher than the calculated TOF from conversion (Fig. 4i). It is noteworthy that the $Cu^{+/2+}$ redox rate with $Pt_SCuO/Al_2O_3$ is consistently superior than the $Cu^{+/2+}$ redox rate with $CuO/Al_2O_3$ across all temperatures. This confirms that the enhanced $Cu^{+/2+}$ redox rate promotes the activity.

## Fast $Cu^{+/2+}$ redox rate leads to higher selectivity of $N_2$

$Pt/Al_2O_3$ is effective at oxidizing ammonia but lacks selectivity to nitrogen gas, especially above 250 °C (Fig. 2b and S9). With $Pt/Al_2O_3$, unfavourable over-oxidation to $N_2O$ and NO takes place on the surface of the $PtO_2$ nanoparticles (Step 2 in the i-SCR)[11,19,33]. Compared to $Pt/Al_2O_3$, the Cu in the $Pt_SCuO/Al_2O_3$ catalyst considerably improves the selectivity to $N_2$ while not affecting the low-temperature activity (Fig. 2a, b), due to the excellent SCR performance of Cu species[34,35]. In

Step 2 of the i-SCR mechanism, NO oxidation on the $Cu^{2+}$ site forms HONO-like species, and the $Cu^{2+}$ is reduced to $Cu^+$[36]. Subsequently, the HONO-like species react further with adsorbed $NH_3$ to generate $N_2$ and $H_2O$, and the $Cu^+$ is then oxidized by $O_2$ to regenerate $Cu^{2+}$, thus completing the redox cycle.

*Operando* DRIFTS confirm that the combination of Pt and Cu sites in $Pt_SCuO/Al_2O_3$ contributes to improved selectivity via the i-SCR (Fig. 5a, b), to achieve the high selectivity to $N_2$ (above 90% over all temperatures)[37]. The bands observed at 1625 and 1,256 $cm^{-1}$ may be assigned to asymmetric and symmetric deformation of ammonia chemisorbed on Lewis acid sites of $Al_2O_3$, respectively[27]. Large amounts of $NH_3$ (1625 and 1256 $cm^{-1}$) are adsorbed on the Lewis acid sites at temperatures below 200 °C. The amount of adsorbed $NH_3$ decreases as the temperature increases, and $-NH_2$ species (evidenced by a peak at 1580 $cm^{-1}$ in the IR spectra) gradually emerge at temperatures above 250 °C[38,39]. It is likely that $NH_3$ dissociatively adsorbs as $-NH_2$ on the surface of the $Pt_SCuO/Al_2O_3$ catalysts. It has been reported that $NH_3$ adsorption on Lewis acid sites is responsible for $NH_3$ oxidation activity, but does not significantly impact $N_2$ selectivity[40]. The presence of Pt in the $Pt_SCuO/Al_2O_3$ catalyst enhances the $Cu^+/Cu^{2+}$ redox recycle, which leads to a higher $NH_3$ oxidation rate. The Lewis acid sites on $Al_2O_3$ observed at 1625 and 1256 $cm^{-1}$ have more pronounced peaks and are more reactive than the Brønsted acid sites (1460 $cm^{-1}$) in the $NH_3$-SCO reaction, since the intensity of the peak at 1460 $cm^{-1}$ hardly changes. It has also been reported that $NH_3$ species adsorbed on Brønsted acid sites can promote the conversion of $NO_x$ formed during $NH_3$ oxidation, thereby improving overall selectivity to $N_2$ through the i-SCR mechanism[41]. $NH_3$ adsorbed on Brønsted acid sites were detected from room temperature to 350 °C on the $Pt_SCuO/Al_2O_3$ catalyst, but are not observed on the $CuO/Al_2O_3$ catalyst. This phenomenon suggests that $NH_3$ species adsorbed on Brønsted acid sites of the $Pt_SCuO/Al_2O_3$ catalyst are more stable than those on $CuO/Al_2O_3$. Presumably the increased stability significantly enhances the selectivity of the $NH_3$–SCO reaction to $N_2$ on the $Pt_SCuO/Al_2O_3$ catalyst. The peak at 1405 $cm^{-1}$ may be assigned to $NH_3$ coordinated to Cu and the peak at 1378 $cm^{-1}$ may be attributed to $NH_3$ coordinated to Pt[36,42]. The presence of Pt in the $Pt_SCuO/Al_2O_3$ catalyst boosts the $Cu^+/Cu^{2+}$ redox cycle, which leads to a higher i-SCR rate and therefore a higher yield to $N_2$.

$NH_3$ emissions are expected to increase in the future and selective catalytic oxidation (SCO) of $NH_3$ to $N_2$ is one of the principle approaches used to eliminate these emissions. The $Pt_SCuO/Al_2O_3$ catalyst reported here is superior to the commercial $Pt/Al_2O_3$ catalysts with respect to both activity and selectivity, achieving full conversion of 5000 ppm $NH_3$ at 250 °C, with selectivity to $N_2$ between 90 and 100%

depending upon the temperature. Based on the moderate operating conditions and the excellent selectivity across a broad temperature range, the catalyst is even suitable for cold start applications such as controlling diesel engine emissions. *Operando* XAFS and time-resolved energy dispersive EXAFS studies were used to show that the enhanced redox rate of the Cu species, induced by the Pt sites in $Pt_SCuO/Al_2O_3$, leads to enhanced activity. *Operando* DRIFTS further demonstrate that the interactions between the Pt and Cu sites contribute to the high selectivity via the i-SCR reaction. Overall, this study illustrates the dynamic changes in the chemical state of the active sites in nanoscale catalysts under relevant reaction conditions, highlighting the importance of *operando* studies to gain a mechanistic understanding of structure–reactivity correlations. The results will help to guide future catalyst design, such as non-noble metal counterparts.

## Methods

### Catalyst preparation

**Synthesis of $Cu/Al_2O_3$.** $\gamma$-$Al_2O_3$ (0.5 g, Johnson Matthey) was dispersed in ethanol (20 mL) with vigorous stirring at room temperature. To the resulting suspension, a solution containing $Cu(NO_3)_2\cdot3H_2O$ (100 mg, 0.4 mmol) in ethanol (5 mL) was slowly added, and stirring was continued for 12 h at room temperature. The reaction mixture was then heated at 40 °C under stirring until all the solvent had evaporated. The remaining solid was heated to 300 °C for 1 h under 15% $H_2$/Ar at a heating rate of 5 °C/min in a tube furnace to afford $Cu/Al_2O_3$.

**Synthesis of $Pt_SCu/Al_2O_3$.** The $Pt_SCu/Al_2O_3$ catalyst was prepared via galvanic replacement between the Cu NPs and $H_2PtCl_6$. $Cu/Al_2O_3$ was dispersed in ethanol (25 mL) under a nitrogen atmosphere, and the resulting suspension was heated at 60 °C for 10 min. A solution of $H_2PtCl_6$ (6 mg) dissolved in ethanol (2 mL) was added slowly to the reaction mixture. After stirring for 6 h at 60 °C, the solution was cooled to room temperature, and then the solid was collected by centrifugation and washed with ethanol (5 × 30 mL). After drying at 40 °C for 24 h, the $Pt_SCu/Al_2O_3$ catalyst was obtained as a grey powder.

**Synthesis of $Pt/Al_2O_3$.** 1 wt% $Pt/Al_2O_3$ was prepared using the wetness impregnation method. $H_2PtCl_6\cdot6H_2O$ (13.4 mg) was dissolved in deionised water and then added into $\gamma$-$Al_2O_3$ (0.5 g) ethanol suspension. The solvent was then removed at 60 °C and the sample was heated at 300 °C for 1 h under 15% $H_2$/Ar at a heating rate of 5 °C/min.

### Ex situ characterisation

**X-ray diffraction (XRD).** XRD patterns were recorded on a StadiP diffractometer (STOE) with a Mo source (K$\alpha$ = 0.7093165 Å). The operating voltage and current were 40 kV and 30 mA, respectively. $2\theta$ in the range of 2–40° were collected with a resolution of 0.015° for each step.

**$H_2$-TPR.** the measurements were performed on an FD-2000 reactor and quantified using an AO2000 analyser. Typically, 50 mg catalyst was placed in a quartz tube and pre-treated in He at 300 °C for 30 min to remove the surface absorbed species. After cooling to room temperature, the sample was heated in 5% $H_2$/$N_2$ (100 mL/min) at a rate of 5 °C/min, then kept at 350 °C for 30 mins.

**TEM.** Aberration-corrected bright field (BF) and annular dark field (ADF) scanning transmission electron microscopy (STEM) was performed on a JEOL (Tokyo, Japan) ARM300CF (E02) operating 300 kV. Simultaneous energy dispersive x-ray (EDX) spectroscopy and aberration-corrected BF/ADF-STEM imaging was performed on a JEOL ARM200CF (E01) operating at 200 kV and equipped with JEOL dual silicon drift detectors at the electron Physical Sciences Imaging Centre (ePSIC) at Diamond Light Source (UK) (DLS). The ARM300CF operated with a convergence semi-angle of 26.2 mrad and BF and ADF collection semi-angles of 0–31.6 and 77.0-209.4, respectively. The ARM200CF

operated with a convergence semi-angle of 23.0 mrad with BF and ADF collection semi-angles of 0–21.9 and 37.5–128.3 respectively. Single-pass EDX spectra were collected with drift correction. Data were acquired and processed using the Gatan Microscopy Suite (a.k.a. Digital Micrograph)[43]. Nanoscale catalyst particles were prepared via a standard preparation route: a small amount (<20 mg) of catalyst powder was dispersed in approximately 5 ml of ethanol, before sonication and drop casting approximately 1 ml of supernatant onto holey carbon coated, gold TEM support grids. Gold was used instead of the more typical copper grid to avoid overlapping fluorescent signals with the sample during EDX mapping. The average particle size was calculated based on more than 100 particles for each sample.

**X-ray absorption fine structure (XAFS).** XAFS of the Pt L$_3$-edge (11.564 keV) and Cu K-edge (8.979 keV) were carried out at the Diamond Light Source (UK) and SPring-8 (Japan). Samples were directly pressed into pellets for fluorescence measurements of the Pt L$_3$-edge and transmission measurements of the Cu K-edge. Pt foil or Cu foil standards were used for energy shift calibration.

XAFS data involved merging three spectra to improve signal quality and were processed using the Demeter software package (including Athena and Artemis). Athena software was used to analyse the XANES data. Artemis software was used to fit the $k^2$-weighted EXAFS data in real space with $3.0\,\text{Å}^{-1} < k < 12.0\,\text{Å}^{-1}$ and $1.0\,\text{Å} < R < 3.3\,\text{Å}$. The calculated amplitude reduction factor $S_0^2$ from the EXAFS analysis of Cu foil was 0.878, which was used as a fixed parameter for EXAFS fitting. The coordination numbers and bond lengths were calculated based on the reported structures from the Crystal open database: Cu (No. 9013014), CuO (No. 1011148), Pt (No. 9008480), and $PtO_2$ (No. 1008935).

### *Operando* Pt L$_3$-edge and Cu K-edge XAFS

*Operando* XAFS experiments were performed at SPring-8 (Japan). 100 mg of pelletised catalysts were measured at 8780–10200 eV for Cu K edge in transmission mode and 11345–12745 eV for Pt L$_3$-edge in fluorescence mode at different temperatures and under various gas atmospheres. Spectra processing was performed with Athena software.

### *Operando* DRIFTS

DRIFTS was performed on a PerkinElmer Frontier FT-IR Spectrometer. The sample was heated in He at 350 °C for 30 min to remove surface contamination. After cooling to room temperature, the sample was exposed to 5000 ppm $NH_3$ or 5%$O_2$/He for 30 minutes, during which spectra were recorded. Then, the sample was heated from 30 to 450 °C with a ramp of 10 °C/min. The spectra were recorded from 4400 to 500 $cm^{-1}$ with a resolution of 2 $cm^{-1}$. Background spectra were recorded in He and subtracted from the sample spectrum for each measurement.

### In situ energy dispersive EXAFS (EDE)

Cu K edge EDE measurements were carried out on the I20-EDE beamline at the Diamond Light Source (UK). For in situ experiments the samples were sieved (125–200 μm) and filled into a quartz tube (5.5 mm). An identical tube filled with $Al_2O_3$ was used as the background. The gas flow was 40 mL/min and spectra were taken when switching the gases. Each condition was run 6 times, and the results were merged for the final spectra.

### In situ near ambient pressure-near edge X-ray absorption fine structure (NAP-NEXAFS) spectroscopy

In situ NAP-NEXAFS experiments were performed on the B07 beamline at the Diamond Light Source (UK)[44]. The X-rays are sourced from a bending magnet (D41) and a plane grating monochromator (PGM) with an energy range from 80 to 2000 eV (soft X-ray range) and flux of $6 \times 10^{10}$ photons/s with 0.1 A ring current using a 111 μm slit and an 80 μm × 200 μm beam spot size. The reaction products were

monitored online using an electron impact mass spectrometer ("PRISMA", PFEIFFER VACUUM GmbH, Asslar (Germany)) connected directly to the main experimental chamber by a leak valve. The pressure in the specimen chamber was precisely controlled at 1 mbar by simultaneous operation of several mass flow controllers for reactive gases and a PID-controlled throttle valve for pumping gas out. Temperature control was achieved by two K-type thermocouples. NEXAFS spectra at Cu L-edge (925–940 eV) were measured in Auger electron yield (AEY) mode.

Measurements were performed under various gas conditions with a total pressure of 1 mbar. In situ experiments employing $Pt_5Cu/Al_2O_3$, $Pt_NCu/Al_2O_3$ and $CuO/Al_2O_3$ were carried out at 200 °C under $NH_3 + O_2$ ($NH_3/O_2$: 1:10) or $NH_3$. For experiments carried out under various gas atmospheres at 200 °C, the sequence of different gas atmospheres follows $NH_3 + O_2$ (ratio: 1:10), $NH_3$, $NO + NH_3$ (ratio: 1:1) and $NH_3 + O_2$ (ratio: 1:10). All spectra were recorded under steady-state conditions.

### Catalytic performance measurements

The performance of the catalysts in the $NH_3$-SCO reaction was evaluated in a fixed-bed flow reactor at a gas flow rate of 100 mL/min, which consists of 5000 ppm $NH_3$, 5 vol% $O_2$, and the He balance. In a typical experiment, 50 mg of catalyst was placed in the reaction tube, and quantification of the products was performed with an online quadrupole mass spectrometer quantitative gas analyser (Hiden Analytical, UK). The reaction was investigated at temperatures ranging from 100 °C to 450 °C. The reaction was kept stable for at least 30 minutes after attaining a steady state at each reaction temperature to detect the MS signals of ($NH_3$ and $O_2$) and products ($N_2$, $N_2O$ and NO).

To obtain the activation energy, 50 mg catalyst powder was immobilized in a fixed-bed flow reactor and a gas flow rate of 100 mL/min, consisting of 5000 ppm $NH_3$, 5 vol% $O_2$, and a He balance was applied. The reaction was investigated at temperatures ranging from 140–180 °C for $Pt_5Cu/Al_2O_3$, 180–220 °C for $Pt/Al_2O_3$ and 200–240 °C for $CuO/Al_2O_3$.

To obtain the reaction order for $NH_3$, the $O_2$ was kept at 5%, while the concentration of $NH_3$ was varied, i.e. 2500, 4000, 5000, 6000 and 7000 ppm. All the tests were performed at 150 °C for $Pt_5Cu/Al_2O_3$, and 220 °C for $CuO/Al_2O_3$.

To obtain the reaction order for $O_2$, the $NH_3$ was kept at 5000 ppm, while the concentration of $O_2$ was varied, i.e. 5, 6, 7, 8, 9 and 10%. All the tests were performed at 150 °C for $Pt_5Cu/Al_2O_3$ and 220 °C for $CuO/Al_2O_3$.

To evaluate the external diffusion limitation, catalyst performance was tested under different flow rates. Different amounts of catalyst powder was immobilized in a fixed-bed flow reactor and the WHSV maintained at 600 mL $NH_3$/g/h. The reaction was investigated at different flow rates of 25, 50, 75, 100, 125 or 150 mL/min at 200 °C and 250 °C.

To evaluate the internal diffusion limitation, the performance of catalysts of differing particle sizes was tested. 50 mg catalyst powder with different particle sizes, including <250 μm, 250–500 μm, and >500 μm, were immobilized in a fixed-bed flow reactor and a gas flow rate of 100 mL/min, consisting of 5000 ppm $NH_3$, 5 vol% $O_2$, and a He balance was applied.

### Data availability

All data generated in this study are provided in the article and Supplementary Information files.

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

## Acknowledgements

We thank the EPSRC for the UKRI Postdoctoral Fellowship (MSCA) to Lu Chen (EP/X022986/1). We thank the EPSRC (EP/S018204/2, EP/ Z001730/1 and EP/Y036220/1) for financial support. We thank EPFL for financial support. We thank the beamline scientists at the SPring-8 (BL14B2, 2022B1758) and Diamond Light Source for the provision of beamtimes (I20-EDE: SP30622-1, B07: SI33257-1) and the scientists in the Electron Physical Science Imaging Centre (EPSIC: MG31922). XAS measurements were performed at BL14B2 of SPring-8 with the approval of the Japan Synchrotron Radiation Research Institute (JASRI) (Proposal No. 2022B1758). We thank the UK Catalysis Hub for the provision of DRIFTS. We thank the Soleil synchrotron (Galaxies beamline: 20221122), the European Synchrotron Radiation Facility (ID-24 beamline: CH-6856 and CH-7062) and the MAX IV (Balder beamline: 20230511) for the provision of beamtimes.

## Author contributions

L.C., X.G., and F.R.W. conceived of the presented idea. L.C., F.R.W. and P.J.D. secured the fundings. L.C. carried out catalysts synthesis and catalytic evaluations, X.G. verified the analytical methods. L.C., X.G., H.A. and F.R.W. carried out the operando XAS study, and L.C. and X.G. analysed the data. L.Z. and Z.F. contributed to the XRD measurements and the XPS measurements. L.C., X.G., Z.W., L.K. and S.H. carried out the in-situ EDE studies. L.C., X.G., X.S. and Z.Y. carried out the NAP-NEXAFS studies with the support from M.V.S., B.K. and G.H. X.G., C.A. and D.H. conducted the TEM measurements. L.C., X.G. performed DRIFTS experiments with the support from D.D. and J.C. L.C., P.J.D. and F.R.W. wrote the paper, and all authors contributed to the final manuscript.

## Competing interests

The authors declare no competing interests.
