## [Transparent Peer Review file · Nature Communications]

Tuning the selectivity of NH₃ oxidation via cooperative electronic interactions between platinum and copper sites

Corresponding Author: Professor Paul Dyson

Version 0:

Reviewer comments:

Reviewer #1

(Remarks to the Author)

This study describes the synthesis of a new type of bimetallic catalyst for selective oxidation of ammonia. With the emergence of ammonia SCR for diesel vehicles, ammonia slip has become an important issue. Its oxidation to benign nitrogen is necessary.

Overall, the paper provides rate/conversion data, comprehensive characterization including Operando measurements, and a proposed mechanism. What is lacking is a complete comparison to application relevant catalysts. This is an important issue since the authors emphasize the finding of a notably better performing catalyst. It is rather surprising that the authors missed a number of key literature references.

The state of art ammonia slip catalyst is the multi-layer monolith catalyst containing an oxidation catalyst (typically Pt) and a SCR catalyst (Fe and/or Cu zeolite). The patent literature describes such catalysts and there are also a few model ASC studies in the open literature. It is further noted that the Pt loading of commercial catalyst is typically less than the 1 wt.% loading that the authors contend. While something can be said for utilizing a single component catalyst rather than dealing with multiple components, it is incumbent on the authors to conduct a comparison. The authors are therefore encouraged to do a more complete literature review and activity comparison.

References that should be consulted include:

Ghosh et al., <https://doi.org/10.1021/acscatal.9b04288>

This paper describes a core-shell catalyst that is very active and selective, and utilizes, in a novel way, the dual layer concept mentioned above. The Pt loading (with alumina support) is commensurate with commercial ASC; i.e. 0.05 wt%. The authors also identify a low loading Pt catalyst modified with silicalite that has a higher activity than the unmodified Pt. The authors should compare their obtained activities to those of Ghosh et al.

Dhillon et al. <https://doi.org/10.1016/j.cattod.2020.01.017> This experimental and modeling study compares a set of dual function monolith catalysts with different distributions of the active materials, including dual layer and mixed. The Pt loading is 1 g/ft³ which translates to a very low Pt wt.%.

Dillon et al. <https://doi.org/10.1039/C8RE00325D> This study describes a dual-layer Pt/Al₂O₃ + Cu/SSZ-13 monolith catalyst.

Marchuk et al. <https://doi.org/10.1021/acscatal.3c05019> This paper provides a comprehensive look at the state of Pt during ammonia oxidation, including the use of Operando methods.

Below are answers to questions raised by the editors:

What are the noteworthy results? Higher activity catalysts with novel bimetallic composition and structure.

Will the work be of significance to the field and related fields? No; the catalyst would appear to be mostly confined to selective oxidation of ammonia.

How does it compare to the established literature? Unclear because of incomplete comparison. See discussion above.

If the work is not original, please provide relevant references. The catalyst appears to be original but again is not compared to complete set of catalysts. See above.

Does the work support the conclusions and claims, or is additional evidence needed? More work is needed to compare the new catalysts to a more relevant low loading ASCs.

Are there any flaws in the data analysis, interpretation and conclusions? A strength of the study is the characterization to support the mechanism proposed.

Do these prohibit publication or require revision? No.

Is the methodology sound? Does the work meet the expected standards in your field? Yes.

Is there enough detail provided in the methods for the work to be reproduced? Yes.

Reviewer #2

(Remarks to the Author)

In this manuscript, the authors report a bi-metallic surficial catalyst (PtSCu/Al₂O₃), which outperforms the commercial Pt/Al₂O₃ catalyst in both activity and selectivity for selective catalytic oxidation (SCO) of NH₃ to N₂. The reaction mechanism is studied by operando XAFS, time-resolved energy dispersive EXAFS, and operando DRIFTS. The results show that Pt atoms in PtSCu/Al₂O₃ enhance the redox properties of Cu species, boost the redox recycle of Cu^{+/2+}, and thus improve the activity and contribute to the high selectivity. This finding is interesting and important. However, after carefully reviewing, many defects were found. Specific comments are listed below.

1. In lines 100-101, the authors reveal the differences between the structures of PtSCuO/Al₂O₃ and PtNCuO/Al₂O₃ by the fitting of EXAFS data. However, the authors did not display the R-factor of the fitting result. And the reasonable range of the Debye-Waller factor is 0.003-0.01, but the fitted results are not within this range. Does the author have a reasonable explanation for this? Otherwise, the fitting results are not credible.

2. It is crucial to determine whether the influence of internal and external diffusion has been excluded in the kinetic experiments. And please state the conditions of the kinetic experiments.

3. In lines 177-179, the authors mention that "Above 250 oC the Cu⁺ feature at 8982 eV disappears and is accompanied by an increase in the white-line intensity at 8996 eV (Figs. 3a, S17 and 18), indicating oxidation of Cu⁺ to Cu²⁺." Please provide pictures that allow for a clearer observation of these changes.

4. Fig. 3i: There are no detailed experimental steps in the whole manuscript, making it difficult for the reader to understand this experiment. And why can this experiment prove the high stability of catalysts?

5. In lines 209-213, the authors claim that "The formation of the Cu⁺ is considered to be a trigger for the NH₃-SCO reaction", is there any more evidence to support this conclusion? What is the role of Pt atoms?

6. In lines 255-257, for the conclusion that "Operando DRIFTS confirms that the combination of Pt and Cu sites in PtSCuO/Al₂O₃ contributes to the improved selectivity via the i-SCR (Fig. 5a,b), to achieve the high selectivity to N₂", please give a more detailed description through literature and experimental data.

7. What is the decomposition temperature of nitrates on Al₂O₃? Generally, nitrate species are more easily generated at low temperatures, and will decompose at high temperatures. Why does it gradually emerge above 250 oC?

8. Some formatting errors:

- a) The figures in SI do not appear in order in the manuscript, which makes reading inconvenient;
- b) In line 83, "NH₃-SCR" should be "NH₃-SCO";
- c) In lines 39-48, pay attention to the subscript;
- d) In line 136, "Table S1" should be Table S2;
- e) What is the difference between the d-f in Figure 3 and Figure S21?
- f) In lines 273-276, the title of the figures does not match the figures.
- g) The catalyst in Figure S25 b is mislabeled.

Version 1:

Reviewer comments:

Reviewer #2

(Remarks to the Author)

Some of my questions have been addressed, but there are still several issues to be resolved.

(1) Response to Comment 2-1: In fact, many papers have problematic EXAFS fitting results. Only for ultra-low temperature systems (such as 10K), the Debye-Waller factor will be lower than 0.003, while only for highly disordered systems (such as molten state sample), the Debye Waller factor will reach 0.02-0.03. The rationality of the Debye factor seriously affects the accuracy of the fitting results of coordination numbers.

(2) Response to Comment 2-2: The conversion rate of NH₃ is controlled below 15%, which makes the measurements in a kinetic region, but this does not mean that the effect of internal and external diffusion is excluded. The authors did not correctly understand the meaning of internal and external diffusion effects.

(3) Response to Comment 2-7: According to the author's explanation, compared with PtsCuO/Al₂O₃ catalysts, CuO/Al₂O₃ has a lower N₂ selectivity above 250 °C, so the peak at 1580 cm⁻¹ should be more pronounced. This does not match the image (Figure 5).

Version 2:

Reviewer comments:

Reviewer #2

(Remarks to the Author)

The manuscript was well revised, all my questions have been addressed, and it is recommended for publication.

Nature Communications manuscript ID: NCOMMS-24-14057-T

Title: Fine-tuning the selectivity of NH₃ oxidation via cooperative electronic interactions between catalytic platinum and copper sites

Note: reviewers' comments appear in black text. Our replies appear in blue text, and the revised text in the manuscript and supporting information appears in green.

Reviewer Comments:

Reviewer #1:

This study describes the synthesis of a new type of bimetallic catalyst for selective oxidation of ammonia. With the emergence of ammonia SCR for diesel vehicles, ammonia slip has become an important issue. Its oxidation to benign nitrogen is necessary.

Overall, the paper provides rate/conversion data, comprehensive characterization including Operando measurements, and a proposed mechanism. What is lacking is a complete comparison to application relevant catalysts. This is an important issue since the authors emphasize the finding of a notably better performing catalyst. It is rather surprising that the authors missed a number of key literature references.

The state of art ammonia slip catalyst is the multi-layer monolith catalyst containing an oxidation catalyst (typically Pt) and a SCR catalyst (Fe and/or Cu zeolite). The patent literature describes such catalysts and there are also a few model ASC studies in the open literature. It is further noted that the Pt loading of commercial catalyst is typically less than the 1 wt.% loading that the authors contend. While something can be said for utilizing a single component catalyst rather than dealing with multiple components, it is incumbent on the authors to conduct a comparison. The authors are therefore encouraged to do a more complete literature review and activity comparison.

References that should be consulted include:

Ghosh et al., <https://doi.org/10.1021/acscatal.9b04288>¹

This paper describes a core-shell catalyst that is very active and selective, and utilizes, in a novel way, the dual layer concept mentioned above. The Pt loading (with alumina support) is commensurate with commercial ASC; i.e. 0.05 wt%. The authors also identify a low loading Pt catalyst modified with silicalite that has a higher activity than the unmodified Pt. The authors should compare their obtained activities to those of Ghosh et al.

Dhillon et al. <https://doi.org/10.1016/j.cattod.2020.01.017>² This experimental and modeling study compares a set of dual function monolith catalysts with

different distributions of the active materials, including dual layer and mixed. The Pt loading is 1 g/ft³ which translates to a very low Pt wt.%.

Dillon et al. <https://doi.org/10.1039/C8RE00325D>³ This study describes a dual-layer Pt/Al₂O₃ + Cu/SSZ-13 monolith catalyst.

Marchuk et al. <https://doi.org/10.1021/acscatal.3c05019>⁴ This paper provides a comprehensive look at the state of Pt during ammonia oxidation, including the use of Operando methods.

We appreciate the reviewer's positive feedback and excellent suggestion to include a more complete literature review and activity comparison. Additional discussion and references (including those proposed by the reviewer) have been included. Additionally, a large number of catalysts from the literature are compared graphically in Fig. 2d and Table S2 has been extended, helping to benchmark the catalysts reported in our manuscript. The new discussion and updated figure and table are provided below:

Additionally, Pt_SCuO/Al₂O₃ outperforms most previously reported catalysts in terms of selectivity to N₂ (Fig. 2d and Table S2).^{10–19,20–23} Compared to benchmark Pt- and Cu-based catalysts, such as the Pt/Al₂O₃@Cu/ZSM-5 core-shell catalyst and Pt/Al₂O₃+Cu/SSZ-13 dual-layer wash-coated monolith catalyst, Pt_SCuO/Al₂O₃ displays a higher activity at low temperatures and maintains high selectivity at high temperatures.^{20–23}

In *operando* XAFS under SCO conditions, indicated that the initial Pt⁴⁺ species in the Pt_SCuO/Al₂O₃, Pt_NCuO/Al₂O₃ and Pt/Al₂O₃ catalysts are reduced to Pt⁰ in the temperature ranges 100-250, 200-300 and 250-300 °C, respectively, and are subsequently re-oxidized to Pt⁴⁺ (Figs. 3b and S17-19). Pt/γ-Al₂O₃ (2 wt%) has been previously studied by operando XAS and was shown to undergo reduction between 230–260 °C,²³ which is similar the catalysts reported herein.

20. Ghosh, R. S., *et al.* *ACS Catal.* **10**, 3604–3617 (2020).
21. Dhillon, P. S., *et al.* *Catal. Today* **360**, 426–434 (2021).
22. Dhillon, P. S., *et al.* *Chem. Eng.* **4**, 1103–1115 (2019).
23. Marchuk, V., *et al.* *ACS Catal.* **14**, 1107–1120 (2024).

Fig. 2 Evaluation of Pt₅CuO/Al₂O₃ in comparison to other catalysts in the NH₃-SCO reaction. **a, b** NH₃ conversion and N₂ selectivity as a function of temperature; **c** activation energy of Pt₅CuO/Al₂O₃, CuO/Al₂O₃ and Pt/Al₂O₃; **d** WSHV of Pt₅CuO/Al₂O₃ and catalysts from the literature (see also Table S2^{10–19,20–23}); **e** stability of Pt₅CuO/Al₂O₃ at 200 °C; **f** activity of M₅CuO/Al₂O₃ catalysts (M = Pt, Ru, Rh, Ag, or Au); **g, h** NH₃ conversion and N₂ selectivity as a function of Pt loading and temperature; **i** NH₃ conversion catalysed by Pt₅CuO/Al₂O₃ with different Pt loadings at 200 °C. Reaction conditions: 50 mg catalyst, 5000 ppm NH₃, 5% O₂ balanced in He, gas flow: 100 mL/min, WSHV= 600 mL NH₃·h⁻¹·g⁻¹.

In the supporting information:

Table S2. Performance of CuO-based bi-functional catalysts evaluated in the NH₃-SCO reaction.

No.	Catalyst	T (°C)	NH ₃ conversion (%)	N ₂ selectivity (%)	WSHV (mL _{NH₃} ·h ⁻¹ ·g ⁻¹)	Ref.
1	Pt ₅ CuO/Al ₂ O ₃ (this work)	200	100	96	600	
2		400	100	98	600	
3	10wt%CuO-Al ₂ O ₃	350	100	93	30	4

4	Pt/Al ₂ O ₃ (1.8 wt.% Pt)	220	100	63	206	5
5	Pt/CuO/Al ₂ O ₃ (1.0 wt.% Pt, 20.0 wt.% Cu)	210	100	88	126	6
6	Pt/CuO/Al ₂ O ₃ (1.0 wt.% Pt, 20.0 wt.% Cu)	250	100	83	-	7
7	Au/Cu/Al ₂ O ₃ (5 wt.% Au/1.0 mol% Cu)	300	100	95	240	8
8	Ag/Cu/Al ₂ O ₃ (7.5 wt.% Ag/2.5 wt.% Cu)	300	100	95	30	9
9	Ag/Cu/Al ₂ O ₃ (7.5 wt.% Ag/2.5 wt.% Cu)	325	100	95	300	10
10	Pt/Al ₂ O ₃ -Cu/ZSM-5 (0.46 wt.% Pt/2.5 wt.% Cu)	250	100	82	-	11
11	30wt%CuO-RuO ₂	210	100	99	75	12
12	4.2 wt% Pd/Al ₂ O ₃	300	100	86	300	13
13	2.6 wt% Pt/Al ₂ O ₃	250	98	43	299	13
14	Pt/Al ₂ O ₃ @Cu/ZSM-5	300	98	94	167	14
15	Pt/Al ₂ O ₃ + Cu/SSZ-13	450	90	80	524	15
16	Pt/Al ₂ O ₃ + Cu/SSZ-13	450	80	90	GHSV: 332k h ⁻¹	16
17	2 wt % Pt/γ-Al ₂ O ₃	230	100	40	1260	17

14. Ghosh, R. S., *et al. ACS Catal.* **10**, 3604–3617 (2020).

15. Dhillon, P. S., *et al. Catal. Today* **360**, 426–434 (2021).

16. Dhillon, P. S., *et al. Chem. Eng.* **4**, 1103–1115 (2019).

17. Marchuk, V., *et al. ACS Catal.* **14**, 1107–1120 (2024).

Reviewer #2:

Comments:

Reviewer #2 (Remarks to the Author):

In this manuscript, the authors report a bi-metallic surficial catalyst (Pt₅Cu/Al₂O₃), which outperforms the commercial Pt/Al₂O₃ catalyst in both activity and selectivity for selective catalytic oxidation (SCO) of NH₃ to N₂. The reaction mechanism is studied by operando XAFS, time-resolved energy dispersive EXAFS, and operando DRIFTS. The results show that Pt atoms in Pt₅Cu/Al₂O₃ enhance the redox properties of Cu species, boost the redox recycle of Cu^{+/2+}, and thus improve the activity and contribute to the high selectivity. This finding is interesting and important. However, after carefully reviewing, many defects were found. Specific comments are listed below.

We greatly appreciate the reviewer's comprehensive and detailed feedback which serve to improve the quality of our work. We have taken into account each point raised, incorporated additional results and discussions accordingly. We believe these efforts will effectively address the reviewer's concerns.

1. In lines 100-101, the authors reveal the differences between the structures of Pt₅CuO/Al₂O₃ and Pt_NCuO/Al₂O₃ by the fitting of EXAFS data. However, the authors did not display the R-factor of the fitting result. And the reasonable range of the Debye-Waller factor is 0.003-0.01, but the fitted results are not within this range. Does the author have a reasonable explanation for this? Otherwise, the fitting results are not credible.

Thank you for raising this issue. Typical values of Debye-Waller factor range from about 0.002 to 0.03 Å². (Scott Calvin, XAFS for everyone, 2013, CRC Press, ISBN 9780429193873, chapter 10). This range is valid for the first shell fitting and for a system which is highly ordered. For second shell fitting (in our studies) slight deviations from these values are quite normal as the disorder increases with bond distance. For example, Feiten et al. reported that EXAFS fits can determine the structures of Pt-containing nanoparticles (*Phys. Chem. Chem. Phys.*, 2020, 22, 18815-18823). They prepared PtCo nanoparticles on carbon, and the coordination number N, the Debye-Waller factor σ^2 and the distance between absorber and scatterer r were fitted. The $\sigma^2(\text{Pt-Co})$ for all six Co-containing samples are drastically reduced with MBA. For PtCoN, PtCoN and AuPtCoN samples, the $\sigma^2(\text{Pt-Co})$ is around 0.0013 Å², which is similar to our results.

Sample	Path	N	$\sigma^2/\text{\AA}^2$	r/\AA
PtCoN bC	Pt-Pt	6.5 ± 1.8	0.0048 ± 0.0015	2.74 ± 0.02
	Pt-Co	3.3 ± 2.1	0.0141 ± 0.0074	2.65 ± 0.11
PtCu_seq/C	Pt-Pt	8.4	0.006 ± 1.10 ⁻⁴	2.75 ± 0.01
	Pt-Cu	2.5	0.015 ± 5.10 ⁻⁴	2.62 ± 0.01
PtCu_seq-at/C	Pt-Pt	8.8	0.006 ± 1.10 ⁻⁴	2.75 ± 0.01
	Pt-Cu	2.3	0.016 ± 5.10 ⁻⁴	2.62 ± 0.01

Pryadchenko et al. (*Applied Catalysis A: General*, 2016, 525, 226–236) prepared PtCu core-shell nanoparticles on carbon. The $\sigma^2(\text{Pt-Cu})$ is around 0.0015 Å². Srabionyan et al. (*Physics of the Solid State*, 2016, 58, 4, 752–762.) also analyzed the PtCu/C catalysts from EXAFS spectroscopy data, and the $\sigma^2(\text{Pt-Cu})$ is 0.0016 Å². These results suggest that the Debye-Waller factor has a reasonable value of 0.013 Å² in the PtCu catalysts.

The R-factor has been included in the Table S1.

EXAFS data fitting was in accordance with literature methods.¹⁻³

Table S1. EXAFS fitting results of the different catalysts.

Sample	Scattering	C.N.	d (Å)	σ^2	E ₀ (eV)	R-factor
Cu foil STD	Cu-Cu	12	2.56			
Cu ₂ O STD	Cu-O	2	1.85			
	Cu-Cu	12	3.01			
CuO STD	Cu-O	4	1.95			
		4	2.88			
	Cu-Cu	4	3.07			
		2	3.16			
	Cu-O	3.42± 0.17	1.95	0.0047± 0.0007	-0.39±0.70	0.013
Pt ₅ CuO/Al ₂ O ₃	Pt-Pt (1)	5.77 ± 1.46	2.69	0.0013 ± 0.0048	2.66 ± 2.53	0.013
	Pt-O (2)	2.00 ± 0.55	1.97			
	Pt-Cu (3)	1.08 ± 0.66	2.67			
Pt _N CuO/Al ₂ O ₃	Cu-O	3.41± 0.16	1.95	0.0029± 0.0068	-0.61±0.62	0.015
	Pt-Pt (1)	4.78± 1.66	2.75	0.011±0.0017	0.85± 2.15	0.019
	Pt-O (2)	1.27± 0.66	1.88			
	Pt-Cu (3)	6.61± 2.40	2.56			

1. Feiten, F. E. *et al. Phys. Chem. Chem. Phys.* **22**, 18815–18823 (2020).
2. Pryadchenko, V. V. *et al. Appl. Catal. A Gen.* **525**, 226–236 (2016).
3. Srabionyan, V. V. *et al. Phys. Solid State*, **58**, 752–762 (2016).

2. It is crucial to determine whether the influence of internal and external diffusion has been excluded in the kinetic experiments. And please state the conditions of the kinetic experiments.

We added the conditions for the kinetic experiments. Note that the conversion of NH₃ is below 15%, so that all the measurements are within the kinetic region to exclude diffusion effects:

Catalytic performance measurements:

To obtain the activation energy, 50 mg catalyst powder was immobilized in a fixed-bed flow reactor and a gas flow rate of 100 mL/min, consisting of 5000 ppm NH₃, 5 vol% O₂, and a He balance was applied. The reaction was investigated at temperatures ranging from 140-180°C for Pt₅Cu/Al₂O₃, 180-220°C for Pt/Al₂O₃ and 200-240°C for CuO/Al₂O₃.

To obtain the reaction order for NH₃, the O₂ was kept at 5%, while the concentration of NH₃ was varied, i.e. 2500, 4000, 5000, 6000 and 7000 ppm. All the tests were performed at 150 °C for Pt₅Cu/Al₂O₃, and 220 °C for

CuO/Al₂O₃.

To obtain the reaction order for O₂, the NH₃ was kept at 5000 ppm, while the concentration of O₂ was varied, i.e. 5, 6, 7, 8, 9 and 10%. All the tests were performed at 150 °C for Pt₅Cu/Al₂O₃ and 220 °C for CuO/Al₂O₃.

3. In lines 177-179, the authors mention that “Above 250 °C the Cu⁺ feature at 8982 eV disappears and is accompanied by an increase in the white-line intensity at 8996 eV (Figs. 3a, S17 and 18), indicating oxidation of Cu⁺ to Cu²⁺.” Please provide pictures that allow for a clearer observation of these changes.

We have added Fig. S18 to highlight the changes of the white-line intensity at 8996 eV.

Above 250 °C the Cu⁺ feature at 8982 eV disappears and is accompanied by an increase in the white-line intensity at 8996 eV (Figs. 3a, S21 and 24), indicative of oxidation of Cu⁺ to Cu²⁺.

Figure S24. Operando Cu K-edge XAFS. Signal intensity of the white-line at 8996 eV in a NH₃/O₂ atmosphere at different temperatures.

4. Fig. 3i: There are no detailed experimental steps in the whole manuscript, making it difficult for the reader to understand this experiment. And why can this experiment prove the high stability of catalysts?

There were many experimental details in the original version of the manuscript, but further details have been added to the Methods to ensure the reproducibility of our work, see below. The reversibility of the oxidation state is considered as an indicator of good stability, we word this cautiously in the manuscript and provide other direct experimental evidence that demonstrate catalyst stability (see Fig. 2e and S10, 100 h continuous stability tests at 200 or 300 °C).

In the NAP-NEXAFS studies, the Cu species in the Pt₅CuO/Al₂O₃ catalyst return to their original redox states after exposure to different gases (Fig. 3i). This switching between Cu⁺ and Cu²⁺ confirms the reversibility of the oxidation state of Cu species in Pt₅Cu/Al₂O₃, as well as the high stability of

the catalyst together with the 100 h stability tests at 200 °C (Figs. 2e).

***In situ* near ambient pressure-near edge X-ray absorption fine structure (NAP-NEXAFS) spectroscopy**

In situ NAP-NEXAFS experiments were performed on the B07 beamline at the Diamond Light Source (UK).⁴¹ The X-rays are sourced from a bending magnet (D41) and a plane grating monochromator (PGM) with an energy range from 80 to 2000 eV (soft X-ray range) and flux of 6×10^{10} photons/s with 0.1 A ring current using a 111 μm slit and an 80 $\mu\text{m} \times 200 \mu\text{m}$ beam spot size. The reaction products were monitored online using an electron impact mass spectrometer ("PRISMA", PFEIFFER VACUUM GmbH, Asslar (Germany)) connected directly to the main experimental chamber by a leak valve. The pressure in the specimen chamber was precisely controlled at 1 mbar by simultaneous operation of several mass flow controllers for reactive gases and a PID-controlled throttle valve for pumping gas out. Temperature control was achieved by two K-type thermocouples. NEXAFS spectra at Cu L-edge (925-940 eV) were measured in Auger electron yield (AEY) mode.

Measurements were performed under various gas conditions with a total pressure of 1 mbar. *In situ* experiments employing $\text{Pt}_s\text{Cu}/\text{Al}_2\text{O}_3$, $\text{Pt}_N\text{Cu}/\text{Al}_2\text{O}_3$ and $\text{CuO}/\text{Al}_2\text{O}_3$ were carried out at 200 °C under NH_3+O_2 (NH_3/O_2 : 1:10) or NH_3 . For experiments carried out under various gas atmospheres at 200 °C, the sequence of different gas atmospheres follows NH_3+O_2 (ratio: 1:10), NH_3 , $\text{NO}+\text{NH}_3$ (ratio: 1:1) and NH_3+O_2 (ratio: 1:10). All spectra were recorded under steady state conditions.

5. In lines 209-213, the authors claim that "The formation of the Cu^+ is considered to be a trigger for the NH_3 -SCO reaction", is there any more evidence to support this conclusion? What is the role of Pt atoms?

In the light-off curve of $\text{Pt}_s\text{Cu}/\text{Al}_2\text{O}_3$, the reaction rates increase rapidly within temperature range 150-200 °C, which corresponds to the temperature for the formation of the Cu^+ . The formation of the Cu^+ is considered to be a trigger for the NH_3 -SCO reaction. See the new discussion and figure below:.

In the light-off curve of $\text{Pt}_s\text{Cu}/\text{Al}_2\text{O}_3$ shown in Fig. S27, the reaction rate increases rapidly from 150 °C, which corresponds to the formation temperature of the $\text{Cu}^+(\text{NH}_3)_2$ species (Figure 3a), so that the formation of the Cu^+ is considered to be a trigger for the NH_3 -SCO reaction.

Figure S27. Light-off curves for Pt₅Cu/Al₂O₃ at different NH₃ concentrations. Reaction conditions: 50 mg catalyst, 1000 or 5000 ppm NH₃, 5% O₂ balanced in He, gas flow: 100 mL/min.

6. In lines 255-257, for the conclusion that “Operando DRIFTS confirms that the combination of Pt and Cu sites in Pt₅CuO/Al₂O₃ contributes to the improved selectivity via the i-SCR (Fig. 5a,b), to achieve the high selectivity to N₂”, please give a more detailed description through literature and experimental data.

As requested, we have expanded the discussion, supported with relevant literature, as shown below:

Operando DRIFTS confirm that the combination of Pt and Cu sites in Pt₅CuO/Al₂O₃ contributes to improved selectivity *via* the i-SCR (Fig. 5a,b), to achieve the high selectivity to N₂ (above 90% over all temperatures).³⁷ The bands observed at 1625 and 1,256 cm⁻¹ may be assigned to asymmetric and symmetric deformation of ammonia chemisorbed on Lewis acid sites of Al₂O₃, respectively.²⁷ Large amounts of NH₃ (1625 and 1,256 cm⁻¹) are adsorbed on the Lewis acid sites at temperatures below 200 °C. The amount of adsorbed NH₃ decreases as the temperature increases, and nitrate species (1580 cm⁻¹) gradually emerge at temperatures above 250 °C. It has been reported that NH₃ adsorption on Lewis acid sites is responsible for NH₃ oxidation activity, but does not significantly impact N₂ selectivity.³⁸ The presence of Pt in the Pt₅CuO/Al₂O₃ catalyst enhances the Cu⁺/Cu²⁺ redox cycle, which leads to a higher NH₃ oxidation rate and therefore a higher formation rate of NO_x. The Lewis acid sites on Al₂O₃ observed at 1625 and 1,256 cm⁻¹ have more pronounced peaks and are more reactive than the Brønsted acid sites (1460 cm⁻¹) in the NH₃-SCO reaction, since the intensity of the peak at 1460 cm⁻¹ hardly changes. It has also been reported that NH₃ species adsorbed on Brønsted acid sites can promote the conversion of NO_x formed during NH₃ oxidation, thereby improving overall selectivity to N₂ through the i-SCR mechanism.³⁹ NH₃ adsorbed on Brønsted acid sites were detected from room

temperature to 350 °C on the Pt₅CuO/Al₂O₃ catalyst, but are not observed on the CuO/Al₂O₃ catalyst. This phenomenon suggests that NH₃ species adsorbed on Brønsted acid sites of the Pt₅CuO/Al₂O₃ catalyst are more stable than those on CuO/Al₂O₃. Presumably the increased stability significantly enhances the selectivity of the NH₃-SCO reaction to N₂ on the Pt₅CuO/Al₂O₃ catalyst. The peak at 1405 cm⁻¹ may be assigned to NH₃ coordinated to Cu and the peak at 1378 cm⁻¹ may be attributed to NH₃ coordinated to Pt.^{36,40} The presence of Pt in the Pt₅CuO/Al₂O₃ catalyst boosts the Cu⁺/Cu²⁺ redox cycle, which leads to a higher i-SCR rate and therefore a higher yield to N₂.

27. Guan, X. *et al. ACS Catal.* **12**, 15207–15217 (2022).

36. Zhang, Q. *et al. Appl. Surf. Sci.* **419**, 733–743 (2017).

37. Burch, R. *et al. J. Catal.* **195**, 217–226 (2000).

38. Lin, M. *et al. ACS Catal.* **9**, 1753–1756 (2019).

39. Jabłońska, M. *et al. J. Catal.* **316**, 36–46 (2014).

40. Yang, J. *et al. Catal. Sci. Technol.* **12**, 1245–1256 (2022).

7. What is the decomposition temperature of nitrates on Al₂O₃? Generally, nitrate species are more easily generated at low temperatures, and will decompose at high temperatures. Why does it gradually emerge above 250 °C?

We understand the concerns from the reviewer about the increased nitrate species at high temperatures. The relative rates of Step 1 (the oxidation of NH₃ to NO) and Step 2 (NO reaction with NH₃ to form N₂) are crucial factors determining the selectivity of the whole process. At temperatures below 250 °C, a higher rate of NO reduction by NH₃ (Step 2) leads to a higher selectivity to N₂. When the temperature is higher than 250 °C, an increased rate of ammonia oxidation to NO (Step 1) results in lower selectivity to N₂.

We added corresponding explanation to the manuscript.

When the temperature is higher than 250 °C, the N₂ selectivity of the Pt_NCuO/Al₂O₃, CuO/Al₂O₃ and Pt/Al₂O₃ catalysts decrease, due to an increase in the rate of ammonia oxidation to NO (Step 1: 4 NH₃ + 5O₂ → 4 NO + 6 H₂O). In comparison, Pt₅CuO/Al₂O₃ consistently maintains >90% selectivity to N₂ even at full conversion (Fig. 2b).

8. Some formatting errors:

We have carefully checked the manuscript and have hopefully fixed all the formatting errors.

a) The figures in SI do not appear in order in the manuscript, which makes reading inconvenient;

We modified the sequence of the figures in the SI as recommended.

b) In line 83, “NH₃-SCR” should be “NH₃-SCO”;

Thank you for spotting the error, we have corrected it.

c) In lines 39-48, pay attention to the subscript;

We corrected the subscript in lines 39-48.

d) In line 136, “Table S1” should be Table S2;

We changed “Table S1” to “Table S2”.

e) What is the difference between the d-f in Figure 3 and Figure S21?

Fig. 3d-f and Figure S21 were the same and Figure S21 has now been removed.

f) In lines 273-276, the title of the figures does not match the figures.

The title of the figures (Fig. 4h,i) has been corrected.

g) The catalyst in Figure S25 b is mislabeled.

The label in the Figure S25 b has been corrected.

Thank you for the reading the manuscript so carefully.

Title: Fine-tuning the selectivity of NH₃ oxidation *via* cooperative electronic interactions between catalytic platinum and copper sites

Note: reviewer's comments appear in black text. Our replies appear in blue text, and the revised text in the manuscript and supporting information is in green.

Reviewer Comments:

Reviewer #2:

Comments:

Reviewer #2 (Remarks to the Author):

Some of my questions have been addressed, but there are still several issues to be resolved.

Thank you for the insightful concerns which have now been fully addressed.

(1) Response to Comment 2-1: In fact, many papers have problematic EXAFS fitting results. Only for ultra-low temperature systems (such as 10K), the Debye-Waller factor will be lower than 0.003, while only for highly disordered systems (such as molten state sample), the Debye Waller factor will reach 0.02-0.03. The rationality of the Debye factor seriously affects the accuracy of the fitting results of coordination numbers.

Thank you for the detailed and helpful explanation for us to better understand the Debye Waller factor. Following this comment, we have re-fitted the EXAFS data and the Debye Waller factors now fall in the range 0.003 and 0.01, which aligns with expectations. With reasonable σ^2 , the new fit also has a smaller error margin.

Pt₅CuO/Al₂O₃ has a larger Pt-Pt coordination number (C.N.) of 5.8 ± 1.5 and smaller Pt-Cu C.N. of 1.1 ± 0.7 compared to Pt_NCuO/Al₂O₃, which has a Pt-Pt C.N. of 4.5 ± 1.4 and a Pt-Cu C.N. of 5.8 ± 1.7 .

Table S1. EXAFS fitting results of the different catalysts.

Sample	Scattering	C.N.	d (Å)	σ^2	E ₀ (eV)	R-factor
Cu foil STD	Cu-Cu	12	2.56			
	Cu-O	2	1.85			
Cu ₂ O STD	Cu-Cu	12	3.01			
	Cu-O	4	1.95			
CuO STD		4	2.88			
	Cu-Cu	4	3.07			
		2	3.16			
Pt ₅ CuO/Al ₂ O ₃	Cu-O	3.42 ± 0.17	1.95	0.005 ± 0.0007	-0.39 ± 0.70	0.013
	Pt-O (1)	2.01 ± 0.55	1.95	0.006 ± 0.004		
	Pt-Pt (2)	5.77 ± 1.46	2.69		1.85 ± 2.30	0.017
	Pt-Cu (3)	1.08 ± 0.66	2.67	0.009 ± 0.002		
Pt _N CuO/Al ₂ O ₃	Cu-O	3.41 ± 0.16	1.95	0.003 ± 0.0007	-0.61 ± 0.62	0.015
	Pt-O (1)	1.75 ± 0.32	1.91	0.005 ± 0.0001		0.022

Pt-Pt (2)	4.48± 1.36	2.79	0.011±0.003	2.10± 2.80
Pt-Cu (3)	5.84± 1.72	2.56		

(2) Response to Comment 2-2: The conversion rate of NH₃ is controlled below 15%, which makes the measurements in a kinetic region, but this does not mean that the effect of internal and external diffusion is excluded. The authors did not correctly understand the meaning of internal and external diffusion effects.

Thank you for your suggestions to consider the meaning of internal and external diffusion effects. To resolve this issue, we performed a series of detailed experiments to ensure that both external and internal diffusion effects were absent under our reaction conditions.

External mass transfer limitations are affected by the volumetric flow rates, which forms a concentration gradient on the surface of the catalysts.

To exclude external diffusion limitations, we varied the gas volumetric flow rate, while keeping the WHSV constant for different amounts of catalyst, confirming that there are no external diffusion limitations when the volumetric flow rate is larger than 75 mL/min (Fig. S10a). In particular, we used 12.5, 25, 37.5, 50 and 62.5 mg of catalysts and kept the WHSV constant at 600 mLNH₃·h⁻¹·g⁻¹.

Internal mass transfer limitations stem from the diffusion of gases within the pores of the catalyst particles. Additionally, we conducted experiments using catalysts with different particle sizes to assess internal diffusion limitations (Fig. S10b). The internal mass transfer limitation is minimized at particle sizes below 500 μm. These findings confirm that the kinetic measurements are not influenced by diffusion limitations.

A series of control experiments were carried out to ensure the influence of internal and external diffusion could be excluded from the kinetic experiments (Fig. S10 a,b).

Figure S10. a Influence of flow rate on NH₃ conversion; b Influence of particle size on NH₃ conversion; c Stability of Pt₅Cu/Al₂O₃ at 300 °C.

To rule out external diffusion limitations, experiments were conducted using different amounts of catalyst at the same WHSV. When the flow rate exceeds 75 mL/min, no external diffusion is present. Since the flow rate used in our experiments are larger than this threshold, there is no external diffusion limitation under the reaction conditions used. Additionally, the catalytic performance with catalysts of differing particle sizes was tested to evaluate potential internal diffusion limitations (Figure S8b). When the particle size is < 500 μm, internal diffusion limitations do not impact on the catalytic performance. In our experiment, the catalyst particle size is smaller than 250 μm, confirming that internal diffusion is not a factor under the given reaction conditions.

Catalytic performance measurements

To evaluate the external diffusion limitation, catalyst performance was tested under different flow rates. Different amounts of catalyst powder was immobilized in a fixed-bed flow reactor and the WHSV maintained at 600 mL NH₃/g/h. The reaction was investigated at different flow rates of 25, 50, 75, 100, 125 or 150 mL/min at 200 °C and 250 °C.

To evaluate the internal diffusion limitation, the performance of catalysts of differing particle sizes was tested. 50 mg catalyst powder with different particle sizes, including <250 μm, 250-500 μm, and >500 μm, were immobilized in a fixed-bed flow reactor and a gas flow rate of 100 mL/min, consisting of 5000 ppm NH₃, 5 vol% O₂, and a He balance was applied.

(3) Response to Comment 2-7: According to the author's explanation, compared with PtsCuO/Al₂O₃ catalysts, CuO/ Al₂O₃ has a lower N₂ selectivity above 250 °C, so the peak at 1580 cm⁻¹ should be more pronounced. This does not match the image (Figure 5).

Thank you for pointing out this discrepancy. After carefully reviewing the literature, we found that the peak at 1580 cm⁻¹ may be assigned to either nitrate species¹ or N-H scissoring of amide -NH₂^{2,3}, depending on the reaction conditions. Under the NH₃-SCR conditions, the peak at 1580 cm⁻¹ is typically assigned to nitrate species, while under our reaction conditions, this peak is more likely attributed to -NH₂ scissoring as nitrate is not present. We have revised the discussion accordingly.

The amount of adsorbed NH₃ decreases as the temperature increases, and -NH₂ species (evidenced by a peak at 1580 cm⁻¹ in the IR spectra) gradually emerge at temperatures above 250 °C.^{38,39} It is likely that NH₃ dissociatively adsorbs as -NH₂ on the surface of the PtsCuO/Al₂O₃ catalysts.

1. Yang, J. et al. Time-resolved in situ DRIFTS study on NH₃ -SCR of NO on a CeO₂/TiO₂ catalyst. *Catal Sci Technol* 12, 1245–1256 (2022).
2. Zhang, L. & He, H. Mechanism of selective catalytic oxidation of ammonia to nitrogen over Ag/Al₂O₃. *J Catal* 268, 18–25 (2009).
3. Cha, B. J. et al. In Situ Spectroscopic Studies of NH₃ Oxidation of Fe-Oxide/Al₂O₃. *ACS Omega* 8, 18064–18073 (2023).